# Health Risk Assessment of Dermal Exposure to Polycyclic Aromatic Hydrocarbons from the Use of Infant Diapers

**DOI:** 10.3390/ijerph192214760

**Published:** 2022-11-10

**Authors:** Alfred Bernard, Vincent Dudler

**Affiliations:** 1Institut de Recherche Expérimentale et Clinique (IREC), Université catholique de Louvain, 74 Avenue Hippocrate, 1200 Bruxelles, Belgium; 2Federal Food Safety and Veterinary Office, Schwarzenburgstrasse 155, 3003 Bern, Switzerland

**Keywords:** diaper, polycyclic aromatic hydrocarbons, skin cancer, neurobehavioral changes

## Abstract

In September 2021, the European Chemicals Agency evaluated a dossier for restricting polycyclic aromatic hydrocarbons (PAHs) in infant diapers and concluded that risks were not demonstrated, because of inconclusive exposure data. To fill this gap, we measured the 16 priority PAHs of the U.S. Environmental Protection Agency in the diaper core of four brands and in the sheets and fastening tapes of six brands of commercially available diapers. Health risks were conservatively assessed by assuming that dermally absorbed PAHs can cause both local (skin cancer) and systemic critical effects (neurobehavioral changes). Total concentrations of PAHs in the diaper core and top sheet, the only significant contributors to skin exposure, averaged 26.5 μg/kg and 66.6 μg/kg, respectively. Excess skin cancer risks and hazard quotients for neurobehavioral effects calculated with the daily dose of total PAHs from the combined diaper core and top sheet averaged 1.44 × 10^−7^ and 1.19 × 10^−2^, respectively. The median daily doses of total PAHs and of its benzo[a]pyrene-equivalent from breast milk estimated worldwide are 171 and 30 times greater than that from the combined diaper core and top sheet, respectively. Altogether, these findings indicate that trace levels of PAHs found in infant diapers are unlikely to pose health risks.

## 1. Introduction

Polycyclic aromatic hydrocarbons (PAHs) are ubiquitous pollutants occurring as complex mixtures throughout the environment. They are formed during the incomplete combustion of fossil fuels and organic matter such as wood or tobacco. They also occur in food as a result of environmental pollution or of some food cooking methods such as grilling and roasting. For nonsmokers, the major sources of exposure are food and, to some extent, polluted air [1,2].

PAHs are organic compounds consisting of multiple fused aromatic rings. The physicochemical properties of PAHs that determine their potential toxicity greatly vary with the number of rings. Two-ring PAHs and, to a lesser extent, three- and four-ring PAHs can partly dissolve in water and are sufficiently volatile to be released in air. By contrast, PAHs with five or more rings have a very low volatility and water solubility. These large PAHs, therefore, are tightly adsorbed onto the surface of solid materials, which reduces their biological accessibility. The strong hydrophobicity of these large PAHs also reduces their solubilization into aqueous liquids while facilitating their transfer across biological membranes and barriers [1,2].

Mixtures of PAHs have long been recognized as potent human carcinogens [3]. In humans, as with animals, the sites of tumors induced by PAHs largely depend on the route of exposure. By inhalation, benzo[a]pyrene (BaP), one of the most potent PAHs, induces only respiratory tract tumors in both humans and rodents, whereas, administered orally in animals, it causes gastro-intestinal tumors. By the dermal route, PAHs caused squamous cell carcinoma of the skin in humans with high occupational exposure [3]. Lifetime carcinogenicity bioassays in several strains of mice have demonstrated that dermally applied BaP induces only skin tumors, which, thus, should be considered as the critical effect of PAHs by the dermal route [1,2,3]. This conclusion is strengthened by the fact that dermal slope factors are more than one order of magnitude greater than oral slope factors. The formation rate of DNA adducts is also orders of magnitude greater in the skin of mice exposed to BaP compared to internal organs [4]. 

Animal studies have also shown that PAHs can cause various systemic effects including developmental, reproductive, and immunological effects [1]. Among these effects, the critical one occurring at the lowest exposure level is the altered neurobehavior observed in rats following gavage or inhalation of BaP during early life. Epidemiological studies among nonsmoking pregnant women mainly exposed to PAHs mixtures via food have reported associations between biomarkers of BaP exposure (benzo[a]pyrene diol epoxide-DNA adducts) and adverse birth outcomes, neurobehavioral effects, and decreased fertility [1].

Recently, public concern arose over exposure to PAHs through the use of disposable infant diapers. A report published in 2019 by the French Agency for Food, Environmental, and Occupational Health and Safety (ANSES), largely echoed in the media, suggested that adverse systemic effects of PAHs, in particular, cancers and neurobehavioral changes, cannot be excluded from the long-term [5]. Therefore, the French agency concluded the need of regulatory actions in order to ensure the safety of diapers. In October 2020, ANSES submitted to the European Chemical Agency (ECHA) a dossier for restricting in infant diapers levels of several hazardous contaminants including PAHs [6]. In September 2021, however, both the ECHA’s Committee for Risk Assessment (RAC) and the Committee for Socio-Economic Analysis (SEAC) considered that the evidence was insufficient to conclude that some chemicals found in diapers may pose a risk to babies [7]. In particular, RAC found that data on the concentrations of some of these substances in diapers were inconclusive. This was especially the case for PAHs as the risk assessment conducted by ANSES relied on the limits of quantification of a poorly sensitive analytical method. In the absence of accurate measurements, RAC was not in a position to completely exclude risks from PAHs in diapers. To fill this gap, we measured PAHs in commercially available infant diapers with an adequate analytical method. We also estimated health risks through a very conservative approach assuming that dermally absorbed PAHs can cause both local (skin cancer) and systemic critical effects (neurobehavioral changes).

## 2. Materials and Methods

### 2.1. Composition of Disposable Diapers

Disposable infant diapers consist of several layers of materials with different functional properties. The most important layer in weight and volume is the diaper core, which is a superabsorbent polymer typically made of sodium polyacrylate granules. In contact with urine, this superabsorbent polymer forms a gel-like material that absorbs up to 30 times its weight in liquid. The top sheet is the inner porous layer of the diaper in direct contact with the baby’s skin. This layer is designed for a rapid transfer of the urine to the diaper core. The back sheet is the water-proof outer layer of the diaper that prevents urine from leaking out of the diaper. Diapers also contain various additional features such as fastening tapes to ensure a good fit [8,9,10].

### 2.2. Measurement of PAHs

This health risk assessment is based on analyses of PAHs performed by the Swiss Federal Institute of Metrology (METAS), which is the reference laboratory in Switzerland for the determination of PAHs in food. The laboratory analyzed the 16 priority PAHs of the U.S. Environmental Protection Agency (U.S. EPA) by gas chromatography–mass spectrometry (GC–MS) using a method adapted from the European standard EN 16619:2015. PAHs were extracted from diapers with n-hexane by liquid-pressure extraction (LPE). For each analysis, about 2 g of material was mixed in a LPE cell with glass beads and 200 µL of a standard solution (mixture of deuterated PAHs). The extraction was conducted in two cycles under the following conditions: pressure 1500 psi, temperature 100 °C, heating time 5 min, static time 10 min, and rinsing volume 10 mL. The extract was concentrated to 5 mL with a rotary evaporator and the analysis was performed by gas chromatography-mass spectrometry (GC-MS). PAHs were measured in the back and top sheets and in the fastening tapes of six international brands of diapers purchased in commercial stores in Switzerland. PAHs were also measured in the diaper core of four of these brands. The diaper core and top sheet had a mean weight of 13 and 4 g, respectively. Results are reported as the mean of three measurements performed on mixtures of three to five randomly selected diapers. The recovery of the PAHs content in the samples was determined from the deuterated labeled PAHs according to the EN 16619:2015 standard, and was always higher than 90%. The limit of quantification (LOQ) is sample-dependent and was measured specifically for each PAH/sample pair. For the 16 substances considered, the range of LOQs and the average were 0.01–2.18 µg/kg and 0.51 µg/kg, respectively. For PAHs that were not quantifiable, we used the LOQ value divided by two to calculate the total PAHs concentrations. In the assessment of skin cancer risks, we calculated the BaP potency equivalent concentrations of individual and total PAHs using the same potency equivalent factors (PEF) as in the ANSES report and our previous study [5,11,12].

### 2.3. Health Risk Assessment

Because the diaper core is not in direct contact with the skin, ingredients in diaper core require urine as an aqueous carrier to reach the skin. The amount of liquid resurfacing back to the skin is, however, strongly limited by the potent absorbency of polyacrylate that allows the achievement of a very low rewetting fraction. In the study of Dey et al. [9], the proportion of urine returning to the top sheet under pressure was estimated on average at 0.46% with a range of 0.32–0.66%. As in our previous study [12], we conservatively used a rewet factor of 1%. By contrast, chemicals in the top sheet may be directly transferred to the skin. For ingredients in the top sheet intended to be applied onto the skin (i.e., top-sheet lotions), Odio et al. [13] estimated that less than 7% is actually transferred to the skin. For ingredients not intended for skin care, their absorption is strongly limited by the fact these ingredients are integrated within the polymeric matrix resin of the sheet. We nevertheless adopted this transfer percentage of 7% as a conservative upper bound estimate of the skin transfer of PAHs from the top sheet. 

As shown previously [12], potential health risks related to infant diaper wearing are expectedly highest in infants aged 0–6 months when the number of diapers used per day and the surface/body weight ratio are highest. We, therefore, focused our risk assessment on this period of age by adopting the same conservative exposure parameters as previously [12] (7.98 diapers per day, body weight of 3.9 kg). The daily dose of PAHs from the diaper core or the top sheet for an infant aged 0–6 months was calculated using the following equations:(1)$$\mathrm{Diaper}\;\mathrm{core}:\;\mathrm{DD}\;=\frac{C\;\times \;W\;\times \;N\;\times \;R\;\times \;A}{\mathrm{SDAF}}$$
(2)$$\mathrm{Top}\;\mathrm{sheet}:\;\mathrm{DD}\;=\frac{\;C\;\times \;W\;\times \;N\;\times \;T\;\times \;A}{\mathrm{SDAF}}$$
where DD is the daily dose (μg/day); C is the concentration of the chemical in the diaper (μg/kg); W is the weight of the diaper layer (kg); N is the number of diapers used per day; T is the fraction transferred to the skin (7%); R is the rewet factor (1%); A is the fraction absorbed by the skin conservatively set at 100%; SDAF is the solvent-dependent adjustment factor incorporated to adjust for the difference in bioaccessibility of PAHs from the diaper core or top sheet when the extraction is performed with a biological fluid at 37 °C or with n-hexane at 100 °C. As there is a close inverse relationship between log Ko/w and water solubility [14] and only a small deviation of 0.4 between log Ko/w and the log Kn-hexane/water of alkyl-aromatic molecules [15], we adopted a SDAF of 100. This adjustment factor of 100 can be considered conservative as it is 20 times lower than the Kow value of the most water-soluble PAH congener (naphthalene) among the 16 priority U.S. EPA PAHs. For the top sheet in direct contact with the skin, we used a lower SDAF to account for the fact that transfer to the skin does not require solubilization in urine. PAHs from the top sheet can indeed directly dissolve in sweat or sebum that is present in variable proportions at the surface of the skin. We adopted a default SDAF of 10 to take into account that the transfer factor of 7% was derived for a product intended for skin care, which presumably should be more easily extracted with n-hexane than PAHs embedded in the top sheet matrix. 

We also estimated the daily dose of PAHs from the diaper core on the basis of the limits of solubility in water of individual PAHs using the following equation:(3)DD =C × R × V
where DD is the daily dose (μg/day); C is the limit of water solubility (μg/L); R is the rewet factor of 1%; V is the daily urine output estimated at 500 mL for a 0–6-months-old infant. The limits of water solubility of individual PAHs were obtained from PubChem https://pubchem.ncbi.nlm.nih.gov (accessed on 1 February 2022).

Excess skin cancer risks of PAHs from the diaper core or the top sheet were calculated by using the following equation: (4)$$\mathrm{ECR}\;=\frac{\;\mathrm{DD}\;\times \;\mathrm{PEF}\;\times \;\mathrm{CSF}\;\times \;T}{S\;\times 70}$$
where ECR is the excess skin cancer risk; DD is the daily dose (μg per day); PEF is the BaP relative potency equivalent factor; CSF is the skin cancer slope factor of 3.5 (μg/cm^2^ per day)^−1^ developed by Knafla et al. [4]; T is the duration of exposure (0.5 years); 70 is the duration of lifetime conventionally set at 70 years; S is the skin surface area in contact with the diaper (234 cm^2^) [16,17]. Unlike our previous study [12], we decided not to incorporate an age-dependent adjustment factor (ADAF) in agreement with the recent opinion of the ECHA risk assessment committee, concluding that there is no need to apply an ADAF for PAHs in addition to the high to low dose extrapolation [18].

Although there are no experimental or epidemiological data to support an extrapolation from ingestion or inhalation to the dermal route of exposure for PAHs, ANSES assumed that PAHs from diapers can cause behavioral changes as observed in rats following oral or inhalation exposure during early life. In case this assumption proves to be correct, we also estimated the risks of neurobehavioral changes by calculating the hazard quotient with the following equation: (5)$$\mathrm{HQ}=\frac{\mathrm{DD}}{\mathrm{TDI}}$$
where HQ is the hazard quotient; DD is the daily dose and TDI is the tolerable daily intake, both expressed in μg/kg body weight and per day, respectively. We assumed a body weight of 3.9 kg for a 0–6-months-old infant and we used as TDI the U.S. EPA reference dose (RfD) of 0.3 μg/kg body weight and per day as used by ANSES [5] and our previous study [12].

## 3. Results

### 3.1. Diaper Core

All investigated PAHs congeners were found in the core of the four tested diaper brands and, depending on the brand, 38% to 81% of them were present in quantifiable amounts (see Appendix A). Of these, there were the two most potent PAHs, BaP and dibenzo[a,h]anthracene (DBA) that were quantified in three and two brands, respectively. The concentrations of BaP and DBA, however, did not exceed 1 μg/kg at the exception of one brand that contained 1.4 μg/kg of DBA. Of note, the concentrations and patterns of PAHs congeners were very consistent across the four brands, being all dominated by naphthalene, which, on average, contributed to 52% of the total PAHs concentrations. Table 1 shows the mean concentrations of PAHs in the diaper core of the four brands and the estimated ECR. The total PAHs concentration in the diaper core averaged 26.5 μg/kg and 1.26 μg/kg when expressed as BaP-equivalent. The ECRs for BaP and DBA were lower than 10^−9^, while the ECR for the total PAHs concentration remained below 10^−8^. As shown in Table 2, the hazard quotient for neurobehavioral effects of total PAHs in the diaper core averaged 2.35 × 10^−4^.

The importance of water solubility in reducing the skin transfer and cancer risk of PAHs from the diaper core clearly emerges from Table 3 showing ECRs calculated with the water solubility limits of individual PAHs. For the most potent PAHs (PEF ≥ 0.1) with water solubility limits lower than 10 μg/L, ECRs were all below 10^−6^. The water solubility limits of the less potent PAHs (PEF 0.01 and 0.001) were much higher, ranging from 135 up to 31,000 μg/L in the case of naphthalene. The ECR of total PAHs reached 2.52 × 10^−5^ and 8.52 × 10^−6^ after exclusion of naphthalene.

### 3.2. Top Sheet

In contrast to the diaper core, concentrations and patterns of PAHs congeners in the top sheet widely varied between the six tested brands (see Appendix A). Depending on the brand, 31% to 50% of PAHs congeners could be quantified. However, only one brand contained a quantifiable amount of the two most potent PAHs congeners (BaP). Table 4 shows the mean concentrations of PAHs in the top sheet of the six tested brands and the associated ECR. The total PAHs concentration averaged 66.6 μg/kg and 3.51 μg/kg when expressed as BaP-equivalent. ECRs for BaP and DBA were lower than 10^−8^ and the ECR for total PAHs remained below 10^−7^. As shown in Table 5, the hazard quotient for neurobehavioral effects of total PAHs in the top sheet was estimated, on average, at 1.27 × 10^−2^.

### 3.3. Diaper Core Plus Top Sheet

Because of the great heterogeneity of top sheet PAHs concentrations between the six diaper brands, we refined our risk assessment by calculating the ECR and hazard coefficients of total PAHs from the combined core and top sheet of the four diaper brands. As shown in Table 6, the ECR ranged from 5.1 × 10^−10^ to 3.1 × 10^−9^ for PAHs in the diaper core and from 5.2 × 10^−8^ to 2.0 × 10^−7^ for PAHs in the top sheet. The ECR for PAHs from the combined diaper core and top sheet averaged 1.44 × 10^−7^ (range, 2.0 × 10^−7^ to 8.60 × 10^−7^), while the hazard quotient for neurobehavioral changes averaged 1.19 × 10^−2^ (range, 9.6 × 10^−3^ to 1.41 × 10^−2^).

### 3.4. Back Sheet and Fastening Tapes

Diaper materials used on the outside of the diaper chassis (e.g., back sheet and fastening tapes) have a very limited direct contact with the skin. Even though there would be some contact with the skin, the contribution of fastening tapes and the back sheet would be totally insignificant when one considers the very low concentrations of PAHs and the patterns of congeners (Appendix A). The two most potent PAHs congeners, BaP and DBA, were mostly undetectable or present in unquantifiable concentrations with the exception of BaP, which was quantified in one brand of fastening tape. Regarding other congeners, only 27% and 29% of them could be quantified in the back sheet and fastening tapes, respectively.

### 3.5. Comparison of the Daily Dose of PAHs from Diaper Core or Top Sheet with That from Breast Milk

Table 7 compares the daily dose of total PAHs and of its BaP-equivalent from the combined diaper core and top sheet with that from breast milk of non-smoking women reported worldwide. The daily dose of total PAHs varied over two orders of magnitude between countries and over more than three orders of magnitude when expressed as BaP-equivalent. These wide variations in the BaP-equivalent daily dose are largely due to BaP and DBA whose concentrations in human milk were undetectable in some studies, while in other studies, they could be quantified with concentrations up to 4.36 μg/kg [19]. The median daily dose of total PAHs and of its BaP-equivalent from breast milk estimated worldwide are 171 and 30 times greater than that from the combined diaper core and top sheet, respectively.

## 4. Discussion

To accurately assess exposure to PAHs from infant diapers, we measured the 16 priority U.S. EPA PAHs in the different layers of commercially available infant diapers in Switzerland. In the diaper core and top sheet, which are the only significant contributors to skin exposure, the proportion of quantifiable PAHs congeners varied depending on the brand between 38% and 81%, and 31% to 50%, respectively. BaP and DBA could be quantified in the diaper core of, respectively, 2 and 3 brands out of the four tested. In the top sheet, only BaP could be quantified in one brand out of the six tested. Of interest, concentrations and patterns of PAHs congeners in the diaper core were remarkably consistent across the brands with a predominance of naphthalene. This consistency suggests a common source of contamination, which is probably linked to ingredients used in the manufacture of the diaper. A possible source of contamination might be the glue used to fix the core as naphthalene is a common contaminant of glues [29]. By contrast, the marked variations in the concentrations and patterns of PAHs congeners in the diaper top sheet suggest the existence of brand-specific sources of contamination that may come from the ingredients, the factory environment, or the widespread environmental pollution by PAHs.

The present study confirms as anticipated [12] that ANSES has largely overestimated cancer risks from PAHs in infant diapers. Our ECR estimates for total PAHs concentration were lower than 10^−8^ for the diaper core and 10^−7^ for the top sheet. These ECRs are more than four orders of magnitude lower than those calculated by ANSES, which, for BaP and DBA alone, exceeded 10^−3^. Interestingly, our study shows that in the case of the diaper core, such a high exceedance is impossible to achieve for a mere physical reason, which is the very poor water solubility of HAPs and especially of BaP and DBA. The ECRs of BaP and DBA in the diaper core calculated at their respective limit of water solubility were indeed in the range of 10^−6^. Furthermore, because of the conservative assumptions made in our study, we probably overestimated the skin exposure to PAHs from the diaper core. Not only did we assume a dermal absorption of 100% but we also conservatively used a SDAF value of 100, which is more than one order of magnitude lower than the lowest Kow value of the 16 priority U.S. EPA PAHs. By incorporating a SDAF value of 100, we assume that the bioavailability of diaper core PAHs is at least 100 times lower when the extraction is performed with urine at 37 °C than with n-hexane at 100 °C. This assumption unavoidably leads to an overestimation of the exposure and health risks of large (≥4 rings) strongly lipophilic PAHs, which have much higher Kow values (log Kow between 5.61 and 6.84) and include the most potent carcinogens (PEF ≥ 0.1). For the top sheet in direct contact with the skin, the transport of PAHs to the skin does not require solubilization in urine.

They can be directly absorbed across the skin after solubilization in sweat or sebum present in variable proportions at the surface of the skin. For top sheet PAHs, we adopted a SDAF of 10 to take into account that the 7% transfer factor was established for substances intended to be delivered to the skin and also that PAHs embedded in the top sheet are presumably more efficiently extracted with n-hexane at 100 °C than with an aqueous solvent at 37 °C. With a skin absorption assumed to be 100%, this SDAF results in a bioavailability of 10%, which can be regarded conservative in regard to the experimental data in the literature. In a human ex vivo skin model, Bourgart et al. [30], for instance, estimated the dermal absorption of unchanged BaP at less than 5% and that of 3-hydroxybenzo[a]pyrene, the metabolite presumably responsible for the neurobehavioral effects of BaP, at less than 0.1% [31]. Moreover, these estimates were made with BaP dissolved in acetone, i.e., under conditions that are known to facilitate the skin absorption of BaP [1,3,4]. Recently, Luo et al. [14] estimated at less than 6% the dermal availability of BaP adsorbed onto indoor dust, which is presumably much more bioavailable than PAHs embedded in the matrix of the diaper top sheet.

Our estimates of skin cancer are in accordance with epidemiological or case report studies that provide no evidence of dermal carcinogenicity of PAHs in infant diapers. In the hypothesis that PAHs from diapers would cause skin cancer, there is no doubt that the critical skin site would be the highly permeable scrotum. It is difficult to believe that such a cancer risk could have passed undetected after more than five decades of widespread use of infant diapers [32]. Squamous cell carcinoma (SCC) is the type of scrotal malignancy that has been reported after high occupational exposure to PAHs. Now, with the early recognition of this hazard and implementation of preventive measures, SCC has become a very rare cancer with a steady incidence through the 20th century. Of note also, the median range of age at SCC diagnosis is 52–57 years, which makes it unlikely that SCC could be initiated during infancy, even though the median SCC latency is close to 30 years [33].

The risk assessment conducted by ANSES was based on both an exposure route and species extrapolation, assuming that dermally absorbed PAH_S_ can cause digestive tract tumors and neurobehavioral changes as evidenced in animals following oral and/or inhalation early exposure. The first assumption about digestive tract cancers is strongly challenged by experiments in rodents showing that dermally applied BaP causes only skin tumors. The second assumption cannot be formally refuted as developmental effects of PAHs have not been investigated in animals by the dermal route. However, if one assumes, as with ANSES, that PAHs absorbed by the skin can cause neurotoxic effects, our findings clearly show that these risks are totally unlikely as the estimated absorbed doses are about two orders of magnitude lower than the U.S. EPA reference dose. This conclusion is indirectly supported by comparing the PAHs daily dose from diapers with that from breast milk, which, in some countries, can be one to three orders of magnitude higher. There is no epidemiological evidence whatsoever associating breastfeeding with increased risks of cancer or neurotoxic effects. On the contrary, breastfeeding is well recognized as protective against a number of diseases or disorders including cancers (e.g., leukemia) and as beneficial to the child’s neurodevelopment, improving the IQ and reducing the risk of behavioral disorders [34,35].

Our study presents some limitations. The first is the lack of experimental data about the bioavailability of PAHs in the diaper core and top sheet. Because extraction was performed with n-hexane, we had to make some conservative assumptions to adjust for the much lower solubility of PAHs in biological fluids compared to n-hexane. One might argue that it would have been more relevant to perform the extraction with artificial urine, the carrier transporting PAHs from the diaper core to the skin. The issue is that given the already very low concentrations found after n-hexane extraction, it is very likely that with an aqueous solvent, most PAHs would have been, if not undetectable, unquantifiable. In addition, extraction with artificial urine would have led to an underestimation of the risks of PAHs in the top sheet, which can be directly transported to the skin without solubilization in an aqueous carrier. Another limitation is the lack of neurotoxicity data for individual PAHs as well as following exposure to PAHs by the dermal route. We conservatively assumed that the 16 U.S. EPA PAHs can cause neurobehavioral changes by the dermal route with the same neurotoxic potency as BaP, which could result in a risk overestimation. Nevertheless, for both the carcinogenic and neurotoxic effects of PAHs, our evaluation has inherent uncertainties due to the extrapolation between species and the possibility of synergistic interactions between PAHs congeners. However, given the very conservative assumptions adopted in our study, we think that these limitations and uncertainties should not change our conclusions.

## 5. Conclusions

We measured the 16 priority U.S. EPA PAHs in the different layers of commercially available infant diapers in Switzerland. In the diaper core and top sheet, the main sources of skin exposure, the proportion of quantifiable PAHs in the different brands varied between 38% and 81%, and 31% to 50%, respectively. The concentrations and patterns of PAHs congeners in the diaper core were remarkably consistent across the brands, which suggests a common source of contamination probably linked to an ingredient used in the manufacture of the diaper. By contrast, both the concentrations and patterns of PAHs congeners greatly varied between brands, which points to different sources of contamination linked to the ingredients and/or the environment. Excess skin cancer risks and hazard quotients for neurobehavioral effects of total PAHs from the combined diaper core and top sheet averaged 1.44 × 10^−7^ and 1.19 × 10^−2^, respectively. The median daily dose of total PAHs and of its BaP-equivalent from breast milk of non-smoking women estimated worldwide are 171 and 30 times greater than that from the combined diaper core and top sheet, respectively. Altogether, these findings indicate that trace levels of PAHs found in infant diapers are unlikely to pose health risks to babies.

## Figures and Tables

**Table 1 ijerph-19-14760-t001:** Excess skin cancer risks of polycyclic aromatic hydrocarbons (PAHs) in the diaper core.

PAHs	Concentration (μg/kg) *	Dose from Diaper Core (μg/day)	PEF	Dose from Diaper Core (μg BaP eqv./day)	Dose from Diaper Core (μg BaP eqv./cm^2^/day)	Excess Skin Cancer Risk
Benzo[a]pyrene	0.55	5.65 × 10^−6^	1	5.65 × 10^−6^	2.42 × 10^−8^	6.04 × 10^−10^
Dibenzo[a,h]anthracene	0.48	4.98 × 10^−6^	1	4.98 × 10^−6^	2.13 × 10^−8^	5.32 × 10^−10^
Benzo[g,h,i]perylene	0.76	7.87 × 10^−6^	0.01	7.87 × 10^−8^	3.37 × 10^−10^	8.41 × 10^−12^
Naphtalene	13.7	1.42 × 10^−4^	0.001	1.42 × 10^−7^	6.07 × 10^−10^	1.52 × 10^−11^
Anthracene	0.81	8.44 × 10^−6^	0.01	8.44 × 10^−8^	3.61 × 10^−10^	9.02 × 10^−12^
Benzo[a]anthracene	0.54	5.56 × 10^−6^	0.1	5.56 × 10^−7^	2.38 × 10^−9^	5.94 × 10^−11^
Indeno[1,2,3-cd]pyrene	0.61	6.29 × 10^−6^	0.1	6.29 × 10^−7^	2.69 × 10^−9^	6.72 × 10^−11^
Chrysene	0.43	4.49 × 10^−6^	0.01	4.49 × 10^−8^	1.92 × 10^−10^	4.80 × 10^−12^
Benzo[b]fluoranthene	0.31	3.23 × 10^−6^	0.1	3.23 × 10^−7^	1.38 × 10^−9^	3.45 × 10^−11^
Benzo[k]fluoranthene	0.51	5.24 × 10^−6^	0.1	5.24 × 10^−7^	2.24 × 10^−9^	5.60 × 10^−11^
Acenaphthene	0.78	8.09 × 10^−6^	0.001	8.09 × 10^−9^	3.46 × 10^−11^	8.65 × 10^−13^
Acenaphthylene	0.35	3.63 × 10^−6^	0.001	3.63 × 10^−9^	1.55 × 10^−11^	3.88 × 10^−13^
Phenanthrene	3.58	3.71 × 10^−5^	0.001	3.71 × 10^−8^	1.59 × 10^−10^	3.97 × 10^−12^
Fluoranthene	1	1.04 × 10^−5^	0.001	1.04 × 10^−8^	4.43 × 10^−11^	1.11 × 10^−12^
Fluorene	1.37	1.42 × 10^−5^	0.001	1.42 × 10^−8^	6.07 × 10^−11^	1.52 × 10^−12^
Pyrene	0.76	7.88 × 10^−6^	0.001	7.88 × 10^−9^	3.37 × 10^−11^	8.42 × 10^−13^
Total PAHs	26.5	1.94 × 10^−4^		1.31 × 10^−5^	5.60 × 10^−8^	1.40 × 10^−9^

* Average of the concentrations measured in the core of four brands of diaper.

**Table 2 ijerph-19-14760-t002:** Assessment of neurobehavioral risks of polycyclic aromatic hydrocarbons (PAHs) in the diaper core.

PAHs	Concentration (μg/kg) *	Dose from Diaper Core (μg/kg/day)	Hazard Quotient
Benzo[a]pyrene	0.545	1.45 × 10^−6^	4.83 × 10^−6^
Dibenzo[a,h]anthracene	0.48	1.28 × 10^−6^	4.26 × 10^−6^
Benzo[g,h,i]perylene	0.76	2.02 × 10^−6^	6.73 × 10^−6^
Naphtalene	13.7	3.64 × 10^−5^	1.21 × 10^−4^
Anthracene	0.81	2.17 × 10^−6^	7.22 × 10^−6^
Benzo(a)anthracene	0.54	1.43 × 10^−6^	4.75 × 10^−6^
Indeno(1,2,3-cd)pyrene	0.61	1.61 × 10^−6^	5.37 × 10^−6^
Chrysene	0.43	1.15 × 10^−6^	3.84 × 10^−6^
Benzo[b]fluoranthene	0.31	8.27 × 10^−7^	2.76 × 10^−6^
Benzo[k]fluoranthene	0.51	1.34 × 10^−6^	4.48 × 10^−6^
Acenaphthene	0.78	2.07 × 10^−6^	6.92 × 10^−6^
Acenaphtylene	0.35	9.31 × 10^−7^	3.10 × 10^−6^
Phenanthrene	3.58	9.52 × 10^−6^	3.17 × 10^−5^
Fluoranthene	1.00	2.66 × 10^−6^	8.87 × 10^−6^
Fluorene	1.37	3.64 × 10^−6^	1.21 × 10^−5^
Pyrene	0.76	2.02 × 10^−6^	6.74 × 10^−6^
Total PAHs	26.5	7.06 × 10^−5^	2.35 × 10^−4^

* Average of the concentrations measured in the core of four brands of diaper.

**Table 3 ijerph-19-14760-t003:** Excess skin cancer risks of polycyclic aromatic hydrocarbons (PAHs) in the diaper core calculated with their respective limit of solubility in water.

PAHs	Number of Rings	Water Solubility (μg/L)	Dose from Diaper Core (μg/day) *	PEF	Dose from Diaper Core (μg BaP eqv./day)	Dose from Diaper Core (μg BaP eqv./cm^2^/day)	Excess Skin Cancer Risk
Benzo[a]pyrene	5	1.62	8.10 × 10^−3^	1	8.10 × 10^−3^	3.46 × 10^−5^	8.65 × 10^−7^
Dibenzo[a,h]anthracene	5	1.66	6.60 × 10^−3^	1	6.60 × 10^−3^	3.55 × 10^−5^	8.87 × 10^−7^
Benzo[g,h,i]perylene	6	0.26	1.30 × 10^−3^	0.01	1.30 × 10^−5^	5.56 × 10^−8^	1.39 × 10^−9^
Naphthalene	2	31,000	1.55 × 10^+2^	0.001	1.55 × 10^−1^	6.62 × 10^−4^	1.66 × 10^−5^
Anthracene	3	43.4	2.17 × 10^−1^	0.01	2.17 × 10^−3^	9.27 × 10^−6^	2.32 × 10^−7^
Benzo[a]anthracene	4	9.4	4.70 × 10^−2^	0.1	4.70 × 10^−3^	2.01 × 10^−5^	5.02 × 10^−7^
Indeno[1,2,3-cd]pyrene	6	0.19	9.50 × 10^−4^	0.1	9.50 × 10^−5^	4.06 × 10^−7^	1.01 × 10^−8^
Chrysene	4	2.0	1.00 × 10^−2^	0.01	1.00 × 10^−4^	4.27 × 10^−7^	1.07 × 10^−8^
Benzo[b]fluoranthene	5	1.5	7.50 × 10^−3^	0.1	7.50 × 10^−4^	8.33 × 10^−5^	8.01 × 10^−8^
Benzo[k]fluoranthene	5	0.8	4.00 × 10^−3^	0.1	4.00 × 10^−4^	8.40 × 10^−5^	4.27 × 10^−8^
Acenaphthene	3	3900	1.95 × 10^1^	0.001	1.95 × 10^1^	8.33 × 10^−5^	2.08 × 10^−6^
Acenaphthylene	3	3930	1.97 × 10^1^	0.001	1.97 × 10^1^	8.40 × 10^−5^	2.10 × 10^−6^
Phenanthrene	3	1300	6.50	0.001	6.50 × 10^−3^	2.78 × 10^−5^	6.94 × 10^−7^
Fluoranthene	4	260	1.30	0.001	1.30 × 10^−3^	5.56 × 10^−6^	1.39 × 10^−7^
Fluorene	3	1690	8.45	0.001	8.45 × 10^−3^	3.61 × 10^−5^	9.03 × 10^−7^
Pyrene	4	135	6.75	0.001	6.75 × 10^−3^	2.88 × 10^−6^	7.21 × 10^−8^
Total PAHs			2.11 × 10^2^		2.36 × 10^−1^	1.01 × 10^−3^	2.52 × 10^−5^
Total PAHs without naphthalene			5.64 × 10^1^		8.07 × 10^−2^	3.45 × 10^−4^	8.62 × 10^−6^

* The daily dose is calculated by assuming a daily urine output of 500 mL for a 0–6-months-old infant.

**Table 4 ijerph-19-14760-t004:** Excess skin cancer risks of polycyclic aromatic hydrocarbons (PAHs) in the top sheet of diapers.

PAHs	Concentration (μg/kg) *	Dose from Top Sheet (μg/day)	PEF	Dose from Top Sheet (μg BaP eqv./day)	Dose from Top Sheet (μg BaP eqv./cm^2^/day)	Excess Skin Cancer Risk
Benzo[a]pyrene	1.6	3.58 × 10^−4^	1	3.58 × 10^−4^	1.53 × 10^−6^	3.82 × 10^−8^
Dibenzo[a,h]anthracene	0.73	1.63 × 10^−4^	1	1.63 × 10^−4^	6.97 × 10^−7^	1.74 × 10^−8^
Benzo[g,h,i]perylene	1.33	2.97 × 10^−4^	0.01	2.97 × 10^−6^	1.27 × 10^−8^	3.17 × 10^−10^
Naphtalene	5.19	1.15 × 10^−3^	0.001	1.15 × 10^−6^	4.96 × 10^−9^	1.24 × 10^−10^
Anthracene	5.61	1.25 × 10^−3^	0.01	1.25 × 10^−5^	5.36 × 10^−8^	1.34 × 10^−9^
Benzo(a)anthracene	6.74	1.51 × 10^−3^	0.1	1.51 × 10^−4^	6.44 × 10^−7^	1.61 × 10^−8^
Indeno(1,2,3-cd)pyrene	0.43	9.61 × 10^−5^	0.1	9.61 × 10^−6^	4.11 × 10^−8^	1.03 × 10^−9^
Chrysene	3.09	6.90 × 10^−4^	0.01	6.90 × 10^−6^	2.95 × 10^−8^	7.38 × 10^−10^
Benzo[b]fluoranthene	1.66	3.71 × 10^−4^	0.1	3.71 × 10^−5^	1.59 × 10^−7^	3.96 × 10^−9^
Benzo[k]fluoranthene	1.54	3.44 × 10^−4^	0.1	3.44 × 10^−5^	1.47 × 10^−7^	3.68 × 10^−9^
Acenaphthene	8.48	1.89 × 10^−3^	0.001	1.89 × 10^−6^	8.10 × 10^−9^	2.02 × 10^−10^
Acenaphtylene	8.16	1.82 × 10^−3^	0.001	1.82 × 10^−6^	7.79 × 10^−9^	1.95 × 10^−10^
Phenanthrene	2.37	5.30 × 10^−4^	0.001	5.30 × 10^−7^	2.26 × 10^−9^	5.66 × 10^−11^
Fluoranthene	2.04	4.56 × 10^−4^	0.001	4.56 × 10^−7^	1.95 × 10^−9^	4.87 × 10^−11^
Fluorene	14.2	3.17 × 10^−3^	0.001	3.17 × 10^−6^	1.36 × 10^−8^	3.30 × 10^−10^
Pyrene	3.4	7.60 × 10^−4^	0.001	7.60 × 10^−7^	3.25 × 10^−9^	8.12 × 10^−11^
Total PAHs	66.6	1.49 × 10^−2^		7.85 × 10^−4^	3.35 × 10^−6^	8.38 × 10^−8^

* Average of the concentrations measured in the top sheet of six brands of diaper.

**Table 5 ijerph-19-14760-t005:** Assessment of neurobehavioral risks of polycyclic aromatic hydrocarbons (PAHs) in the top sheet of diapers.

PAHs	Concentration (μg/kg) *	Dose from Top Sheet (μg/kg/day)	Hazard Quotient
Benzo[a]pyrene	1.6	9.17 × 10^−5^	3.06 × 10^−4^
Dibenzo[a,h]anthracene	0.73	4.18 × 10^−5^	1.39 × 10^−4^
Benzo[g,h,i]perylene	1.33	7.62 × 10^−5^	2.54 × 10^−4^
Naphtalene	5.19	2.97 × 10^−4^	9.91 × 10^−4^
Anthracene	5.61	3.21 × 10^−4^	1.07 × 10^−3^
Benzo(a)anthracene	6.74	3.86 × 10^−4^	1.29 × 10^−3^
Indeno(1,2,3-cd)pyrene	0.43	2.46 × 10^−5^	8.21 × 10^−5^
Chrysene	3.09	1.77 × 10^−4^	5.90 × 10^−4^
Benzo[b]fluoranthene	1.66	9.51 × 10^−5^	3.17 × 10^−4^
Benzo[k]fluoranthene	1.54	8.82 × 10^−5^	2.94 × 10^−4^
Acenaphthene	8.48	4.86 × 10^−4^	1.62 × 10^−3^
Acenaphtylene	8.16	4.68 × 10^−4^	1.56 × 10^−3^
Phenanthrene	2.37	1.36 × 10^−4^	4.53 × 10^−4^
Fluoranthene	2.04	1.17 × 10^−4^	3.90 × 10^−4^
Fluorene	14.2	8.14 × 10^−4^	2.71 × 10^−3^
Pyrene	3.40	1.95 × 10^−4^	6.49 × 10^−4^
Total PAHs	66.6	1.49 × 10^−2^	1.27 × 10^−2^

* Average of the concentrations measured in the top sheet of six brands of diaper.

**Table 6 ijerph-19-14760-t006:** Excess skin cancer and neurobehavioral risks of polycyclic aromatic hydrocarbons (PAHs) in the core and the top sheet of four brands of diaper, separately or combined.

Brands	Part of Diaper	Total PAHs Concentration (μg/kg)	Dose from Diaper (μg BaP eqv./cm^2^/day)	Excess Skin Cancer Risk	Dose from Diaper (μg/kg/day)	Hazard Quotient
A	Top sheet	57.2	8.1 × 10^−6^	2.0 × 10^−7^	3.28 × 10^−3^	1.09 × 10^−2^
	Core	17.8	3.0 × 10^−8^	7.6 ×10^−10^	4.74 × 10^−6^	1.58 × 10^−5^
	Top sheet + core		8.0 × 10^−6^	2.0 × 10^−7^	3.29 × 10^−3^	1.10 × 10^−2^
B	Top sheet	65.6	2.1 × 10^−6^	5.2 × 10^−8^	3,74 × 10^−3^	1.25 × 10^−2^
	Core	34.9	4.7 × 10^−8^	1.2 × 10^−9^	9.28 × 10^−5^	3.09 × 10^−4^
	Top sheet + core		2.1 × 10^−6^	1.7 × 10^−7^	3.84 × 10^−3^	1.28 × 10^−2^
C	Top sheet	72.3	4.5 × 10^−6^	1.1 × 10^−7^	4.13 × 10^−3^	1.36 × 10^−2^
	Core	30.2	1.3 × 10^−7^	3.1 × 10^−9^	8.03 × 10^−5^	2.68 × 10^−4^
	Top sheet + core		4.7 × 10^−6^	1.2 × 10^−7^	4.21 × 10^−3^	1.40 × 10^−2^
D	Top sheet	49.4	3.4 × 10^−6^	8.5 × 10^−8^	2.82 × 10^−3^	9.40 × 10^−3^
	Core	22.5	2.0 × 10^−8^	5.1 × 10^−10^	6.00 × 10^−5^	2.00 × 10^−4^
	Top sheet + core		3.4 × 10^−6^	8.6 × 10^−8^	2.88 × 10^−3^	9.60 × 10^−3^
All	Top sheet + core		3.79 × 10^−6^	1.44 × 10^−7^	3.56 × 10^−3^	1.19 × 10^−2^

**Table 7 ijerph-19-14760-t007:** Comparison of the BaP-equivalent daily dose of PAHs from breast milk of non-smoking women with that from diapers.

Country	Authors	Dose of PAHs from Breast Milk *		Breast Milk/Diaper Dose Ratio	
		μg/kg/Day	μg BaP eqv./kg/Day	μg/kg/Day	μg BaP eqv./kg/Day
Colombia	Torres-Moreno et al., 2022 [20]	0.62	6.13 × 10^−3^	224	29
Portugal	Oliveira et al., 2020 [21]	1.28	6.48 × 10^−2^	462	304
USA	Acharya et al., 2019 [22]	0.52	1.86 × 10^−2^	188	87
Ghana	Asamoha et al., 2019 [23]	0.43	6.14 × 10^−3^	155	29
Italy	Santonicola et al., 2017 [19]	16.1	1.00	5812	4695
Czech Republic	Pulkrabova et al., 2016 [24]	0.093	9.53 × 10^−4^	34	4.5
Turkey	Cok et al., 2012 [25]	0.30	2.18 × 10^−3^	108	10
USA	Kim et al., 2008 [26]	0.084	2.90 × 10^−4^	30	1.3
Italy	Zanieri et al., 2007 [27]	2.58	1.16 × 10^−2^	931	54
Japan	Kishikawa et al., 2003 [28]	0.11	6.75 × 10^−3^	40	32

* Daily dose from breast milk was calculated for an infant of 5 kg of body weight fed daily with 700 mL of maternal milk containing 25 g/L of lipids. The daily dose from the diaper is the daily dose of PAHs from combined diaper core and top sheet.

## Data Availability

Concentrations of PAHs in the different parts of diapers are provided in the Appendix A.

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
