# Peer review of "Health Risk Assessment of Dermal Exposure to Polycyclic Aromatic Hydrocarbons from the Use of Infant Diapers"

_ijerph, 2022, doi:10.3390/ijerph192214760_

Round 1
Reviewer 1 Report
The manuscript, entitled "Health Risk Assessment of Dermal Exposure to Polycyclic Aromatic Hydrocarbons from The Use of Infant Diapers" (ijerph-1993247), examined the PAHs contents in commercially infant diapers and assessed their health risks. I think that the paper is well-written and professionally presented, except for a few grammar errors. I enjoyed reading this manuscript. The design is adequate to meet the project objectives, and the conclusions drawn were derived from the data presented. The findings could provide further new insights into the health risk of using infant diapers. I believe that this paper can be considered for publication in the journal after some minor modifications.
Specific comments:
(1) In section 2.2, I suggest the authors add more descriptions in detail on the procedures and parameters of PAHs measurement.
(2) Line 54-55, “Animal studies have also shown that…”, please add the related references.
(3) Line 62, “…over the exposure…”, delete “the”.
(4) Line 66, change “…over…” by “…from…”.
(5) In section 3.1, please add the PAHs recovery, which is an important information for PAH chemical analysis.
(6) Line 248, replace “unsignificant” with “insignificant”.
Author Response
The manuscript, entitled "Health Risk Assessment of Dermal Exposure to Polycyclic Aromatic Hydrocarbons from The Use of Infant Diapers" (ijerph-1993247), examined the PAHs contents in commercially infant diapers and assessed their health risks. I think that the paper is well-written and professionally presented, except for a few grammar errors. I enjoyed reading this manuscript. The design is adequate to meet the project objectives, and the conclusions drawn were derived from the data presented. The findings could provide further new insights into the health risk of using infant diapers. I believe that this paper can be considered for publication in the journal after some minor modifications.
specific comments:
(1) In section 2.2, I suggest the authors add more descriptions in detail on the procedures and parameters of PAHs measurement.
Re: The description of the procedure for measuring PAHs has been expanded by adding the following details about the extraction procedure, the analytical recovery and the limits of quantification:
PAHs were extracted from diapers with n-hexane by liquid pressure extraction (LPE). For each analysis, about 2 g of material was mixed in a LPE cell with glass beads and 200 µl of a standard solution (mixture of deuterated PAHs). The extraction was conducted in two cycles under the following conditions: pressure 1500 psi, temperature 100°C, heating time 5 min, static time 10 min, rinsing volume 10 ml. The extract was concentrated to 5 ml with a rotary evaporator and the analysis was performed by gas chromatography-mass spectrometry (GC-MS).
The recovery of the PAHs content in the samples was determined from the deuterated labeled PAHs according to the EN 16619:2015 standard, and was always higher than 90%. The limit of quantification (LOQ) is sample dependent and was measured specifically for each PAH/sample pair. For the 16 substances considered, the range of LOQs and the average were 0.01-2.18 µg/kg and 0.51 µg/kg, respectively.
(2) Line 54-55, “Animal studies have also shown that…”, please add the related references.
Re: OK. The adequate reference [1] has been added.
(3) Line 62, “…over the exposure…”, delete “the”.
Re: Thank you. “The” has been deleted
(4) Line 66, change “…over…” by “…from…”.
Re: Thank you. This has been changed.
(5) In section 3.1, please add the PAHs recovery, which is an important information for PAH chemical analysis.
Re: As indicated above (1), data about recovery of PAHs have been added in section 2.2 Measurement of PAHs.
(6) Line 248, replace “unsignificant” with “insignificant”.
Re: Thank you. This has been corrected.
Reviewer 2 Report
The design and results of this study are well written. There are, however, some points needed for clarification. Comments and suggestions are as follows:
1. Lines 170-171, T is the duration of exposure (0.5 years), and 70 years are supposed to the lifetime. Please revise the sentence accordingly. By the way, why is the exposure duration 0.5 years? Most infants use diapers 2-3 years as usual. The authors may want to answer that.
2. ECR and HQ for neurobehavioral effects are given as 1.4 X 10^-7 and 1.19 X 10^-2 in Abstract as well as Conclusions, respectively. From the tables as many as 7, these numbers do not seem to be derived from any one of them. Please check and possibly reduce the number of tables for better reading.
3. ECR for naphthalene in Table 3 is higher than 10^-6. Is that of concern? Yes or no, why?
4. Table 6, Please use A, B, C and D to distinguish the four brands. The two rows in the bottom are both titled as "Top sheet," which confuse readers.
5. Please define PEF in the first table (Table 1).
6. Line 104, what are the LOQ values? It may not be necessary to report each one, but a range of the LOQs should be given in the text.
Author Response
The design and results of this study are well written. There are, however, some points needed for clarification. Comments and suggestions are as follows:
- Lines 170-171, T is the duration of exposure (0.5 years), and 70 years are supposed to the lifetime. Please revise the sentence accordingly. By the way, why is the exposure duration 0.5 years? Most infants use diapers 2-3 years as usual. The authors may want to answer that.
Re: Thank you. This has been corrected line 180 as follows: ”70 is the duration of lifetime conventionally set at 70 years”. As done in our previous study (reference 13) and explained in our paper (line 133) we focused on the 6 first months of life because potential health risks are highest at this period when the number of diapers used per day and the surface/body weight ratio are the highest. Our risk estimates should not differ by more than a factor of 2-3 because between birth and the age of 2-3 years the number of diapers used per day decreases by almost 50%, the surface/body weight ratio decreases by about 30 % and with the skin maturation the percutaneous absorption of PAHs conservatively assumed at 100% most probably also decreases. A difference of 2-3 in our risk estimates should not change our conclusions because of the very conservative assumptions made in our study.
- ECR and HQ for neurobehavioral effects are given as 1.4 X 10^-7 and 1.19 X 10^-2 in Abstract as well as Conclusions, respectively. From the tables as many as 7, these numbers do not seem to be derived from any one of them. Please check and possibly reduce the number of tables for better reading.
Re: This is right. These ECR and HQ mean values were given only in the text and they did not appear in the Tables. This has been corrected. These numbers have been adder in the bottom of Table 6.
- ECR for naphthalene in Table 3 is higher than 10^-6. Is that of concern? Yes or no, why?
Re: No of course because this cancer risk estimate is not based on actual concentrations of naphthalene measured in the diaper core but on the limit of water solubility of naphthalene. As explained in our paper, the objective of this table is to demonstrate that in the case of diaper core – the main diaper layer in weight and volume – cancer risk as high as 10-3 estimated by ANSES are impossible to achieve because of the poor water solubility of PAHs and especially of the PAHs with the highest PEF. This table also underscores the importance of assessing cancer risks separately for the diaper core and the top sheet as PAHs in the latter do not require urine as carrier to reach the skin.
- Table 6, Please use A, B, C and D to distinguish the four brands. The two rows in the bottom are both titled as "Top sheet," which confuse readers.
Re: This has been done. We now distinguish the brands with the letters A, B, C and D. The two rows in the bottom for top sheets have been deleted.
- Please define PEF in the first table (Table 1).
Re: This has been done in the footnote of the Table 1. PEF, potency equivalent factor.
- Line 104, what are the LOQ values? It may not be necessary to report each one, but a range of the LOQs should be given in the text.
Re: Thank you for this suggestion. As stated above in response to the same request of the first reviewer, the following section has been added in Materials and Methods:
“The recovery of the PAHs content in the samples was determined from the deuterated labeled PAHs according to the EN 16619:2015 standard, and was always higher than 90%. The limit of quantification (LOQ) is sample dependent and was measured specifically for each PAH/sample pair. For the 16 substances considered, the range of LOQs and the average were 0.01-2.18 µg/kg and 0.51 µg/kg, respectively.”